# Synthesis with Immobilized Lipases and Downstream Processing of Ascorbyl Palmitate

**DOI:** 10.3390/molecules24183227

**Published:** 2019-09-05

**Authors:** Carolina Tufiño, Claudia Bernal, Carminna Ottone, Oscar Romero, Andrés Illanes, Lorena Wilson

**Affiliations:** 1School of Biochemical Engineering, Pontificia Universidad Católica de Valparaíso, Avenida Brasil, Valparaíso 2085, Chile (C.T.) (C.O.) (O.R.) (A.I.); 2Instituto de Investigación Multidisciplinario en Ciencia y Tecnología, Universidad de La Serena, Raúl Bitrán, La Serena 1305, Chile

**Keywords:** ascorbyl palmitate, lipase, enzymatic synthesis, palmitate, antioxidants

## Abstract

Ascorbyl palmitate is a fatty acid ester endowed with antioxidant properties, used as a food additive and cosmetic ingredient, which is presently produced by chemical synthesis. Ascorbyl palmitate was synthesized from ascorbic acid and palmitic acid with a *Pseudomonas stutzeri* lipase immobilized on octyl silica, and also with the commercial immobilized lipase Novozym 435. The latter was selected for optimizing the reaction conditions because of its high reactivity and stability in the solvent 2-methyl-2-butanol used as reaction medium. The reaction of the synthesis was studied considering temperature and molar ratio of substrates as variables and synthesis yield as response parameter. The highest yield in the synthesis of ascorbyl palmitate was 81%, obtained at 55 °C and an ascorbic acid to palmitic acid molar ratio of 1:8, both variables having a strong effect on yield. The synthesized ascorbyl palmitate was purified to 94.4%, with a purification yield of 84.2%. The use of generally recognized as safe (GRAS) certified solvents with a polarity suitable for the solubilization of the compounds made the process a viable alternative for the synthesis and downstream processing of ascorbyl palmitate.

## 1. Introduction

Antioxidants are molecules that delay, prevent or suppress the oxidative damage of molecules prone to oxidation, such us lipids and fats [1,2]. Antioxidants can be obtained by chemical synthesis, as in the case of butyl hydroxyanisole (BHA) and butyl hydroxytoluene (BHT), or by extraction from natural sources, as in the case of ascorbic acid and tocopherols [3]. However, in recent years, toxicological research in animals showed that the high consumption of antioxidants from synthetic origin produced carcinogenic effects [4]. Ascorbic acid (AA) is a natural antioxidant of technological relevance, which is considered a safe ingredient in different products [5,6].

AA is a water-soluble vitamin with a high antioxidant activity. However, its hydrophilic character hinders its use in the stabilization of fats and oils [3]. To overcome this problem, AA can be esterified with different fatty acids to produce the corresponding ascorbyl esters and obtain new compounds with antioxidant as well as surfactant properties [7]. The most used substrates for these reactions are the long chain fatty acids since as the chain length increases the non-polar character increases, which is the case of palmitic acid (PA) in the synthesis of ascorbyl palmitate [8,9]. Ascorbyl palmitate (AsPa) is one of the most important ascorbic acid esters since it is an amphipathic antioxidant that has several applications in foods, pharmaceuticals and cosmetics [2].

The industrial production of AsPa has two important aspects to consider: the selection of the type of catalysis and the downstream process to achieve the required quality of the final product. Industrially, the preferred system has been the chemical synthesis using concentrated sulfuric acid or hydrofluoric acid as catalyst. This is a process with high energy expenditure that results in the formation of by-products that hinder its purification [10]. More recently, the enzymatic synthesis using lipases as biocatalyst has been considered as an alternative for the synthesis of AsPa. In fact, enzyme biocatalysis has several advantages such us less byproduct formation and eco-friendly reaction conditions more in compliance with the principles of green chemistry. Lipases can catalyze the synthesis of AsPA by direct esterification or transesterification, using palmitic acid or palmitic esters respectively [11]. From an atom economy point of view, direct esterification seems to be a more attractive synthetic strategy; however, it is necessary to avoid water accumulation to shift the thermodynamic equilibrium in favor of hydrolisis [12,13]. Several approaches have been used to solve this problem, namely, the addition of molecular sieves [14], the use of ultrasound [15] and the immobilization of the enzyme on hydrophobic supports [16].

The immobilization of lipases on hydrophobic supports by interfacial adsorption allows fixing the open conformation on a solid phase, representing a simple approach to obtein very active and stable catalysts [17]. Moreover, the immobilized lipase may exhibit improved properties, in terms of activity, selectivity, stability and resistant to inhibitors, among others [18].

Another major problem concerning enzymatic synthesis is that a reaction medium in which substrate solubility and enzyme activity are well balanced is hard to attain. Selection of reaction conditions is a key aspect in the enzymatic acylation of polar compounds. The lipase-catalyzed synthesis of fatty acid ascorbyl esters has been thoroughly studied in different reaction media, including organic solvents, ionic liquids and mixtures of cosolvents. Preferred solvents for ascorbyl esters synthesis are 2-methyl-2-butanol (tert-amyl alcohol) and butanol [2].

The first studies published on the enzymatic synthesis of AsPa used the lipase from *Candida antarctica*, where the aim was a comparison between direct esterification using palmitic acid or transesterification with methyl palmitate, higher yield being obtained by transesterification [11]. Bradoo, et al. [19] used the lipase from *Bacillus stearothermophilus* obtaining an AsPa yield of 97%, at 60 °C in the presence of hexane. Later, Viklund, et al. [9] studied the influence of the AA:PA molar ratio, finding that a higher PA concentration favored AsPa yields; however, the range of molar ratios studied was very narrow. Lerin, et al. [8] evaluated a broader molar range confirming the results and obtaining an AsPa yield of 67% with Novozym 435 at 70 °C in the presence of tert-butanol at an AA:PA molar ratio of 1:9 (the lower value tested). Lipases other than Novozym 435 have been seldom used for AsPa synthesis, the *Pseudomonas stutzeri* lipase being used for first time by Santibañez, et al. [20]. Lipases from *Pseudomonas* are widely used in industrial application, due to their ability to catalyze many reactions, stability in organic solvents and low cost [21]. Santibañez, et al. [20] used the enzyme immobilized in octyl-agarose obtaining a synthesis conversion of 57% at an AA:PA molar ratio of 1:5, using tert-butanol as solvent. Sharma, et al. [22] evaluated the esterification of AA with PA, using 1 M AA and 2.5 M PA in dimethyl sulfoxide (DMSO) at 75 °C for 18 h, reaching 80% of synthesis conversion using the commercial lipase Lipolase 100 L. Recently, synthesis of AsPa in tert-butyl alcohol was carried out using immobilized lipase from *Candida antarctica* B at optimized reaction conditions obtaining in conversion of 90% [23]. Despite several studies have been published in this topic, there are no reports about the combined effect of higher AA:PA molar ratios and temperature, neither the evaluation of the influence of more polar solvents on the enzymatic performance in AsPa production, which could lead to high synthesis yield under softer reaction conditions.

Downstream operations in the process of AsPa production have been scarcely explored. The chemical route for AsPa synthesis poses more difficulty to downstream because of the presence of several byproducts of the reaction that contaminate the product and must be removed [2,10]. In the enzymatic synthesis of AsPa, purification is also required but to a lesser extent, and fewer and less aggressive operations are required; in this case the task is merely the separation of AsPa from the residual substrates. At lab scale two methods of AsPa purification are customary: semi-preparative high-performance liquid chromatography (HPLC) [24,25,26], which requires sophisticated and costly equipment, and solvent extraction [11,24,27], which has also been used for the purification of ascorbyl oleate [9] and erythorbyl laurate [28]; however, this process has not been validated and/or optimized with respect to purification yield and purity of the product.

The aim of this work is the evaluation of the enzymatic synthesis of AsPa from the standpoint of the biocatalyst and reaction conditions and their effect on downstream parameters, namely, purity of the final product and purification yield. This works makes a comparison of the performance of the *P. stutzeri* lipase immobilized on hydrophobic silica and the commercial immobilized lipase Novozym 435, selecting the best catalyst in terms of AsPa synthesis yield, using three solvents (acetone, acetonitrile and 2-methyl-2-butanol) as reaction media, considering temperature and AA:PA molar ratio as variables. Then, a donwnstream process was evaluated in terms of purity and purification yield of AsPa. This work is a first report for an integrated process of synthesis and downstream processing for AsPa production, representing a useful tool for scaling up the process to industrial level.

## 2. Results and Discussion

### 2.1. Biocatalyst Characterization

*P. stutzeri* lipase immobilized in octyl silica (PS-octyl-silica) was obtained according to a previously reported procedure [29]. Hydrolytic specific activity of PS-octyl-silica was 70.44 IU/g_biocatalyst_ and the IY_a_ and IY_p_ of PS-octyl-silica were 51% and 92%, respectively, which can be considered normal for this kind of enzyme. On the other hand, Novozym 435 was also characterized, determining a hydrolytic specific activity for of 34.61 IU/g_biocatalyst_.

### 2.2. Stability in the Presence of Solvents

Since AsPa synthesis is carried out in the presence of a high concentration of organic solvent, the stability of the PS-octyl-silica and Novozym 435 biocatalyst was studied by incubation under non-reactive conditions with the three solvents at 45 °C (Figure 1). The residual activity for both enzymes, after incubation for 144 h, is presented in Table 1.

Several authors have reported that thermal stability is dependent on the solvent, on the immobilization methodology (support and chemical interactions) and on the lipase source [30]. PS-octyl-silica lipase presented quite different deactivation kinetics in the three solvents (Figure 1A): in acetone the enzyme activity was kept constant at 84% for 144 h, while in the presence of acetonitrile the enzyme evidenced hyperactivation (values of residual activity higher than initial) during such a period. The phenomenon of hyperactivation of immobilized lipases in the presence of solvents has been well documented [31]. In contrast, the residual activity of the biocatalyst in 2M2B decreased to 49% after 72 h of incubation. Such different behavior can be due to the solvent polarity, since both acetonitrile and acetone are polar, while 2M2B is a non-polar solvent (dielectric constants are 37, 20.7 and 5.8 for acetonitrile, acetone and 2M2B, respectively). Since octyl-silica support is hydrophobically funcionalyzed, this may promote the contact of the immobilized enzyme with 2M2B favoring inactivation [32,33].

The behavior of Novozym 435 was completely different (Figure 1B): enzyme activity decreased rapidly when incubated with acetone and acetonitrile (less than 40% residual activity after 24 h), and later on it remained constant until 144 h. On the hand, when Novozym 435 was incubated with 2M2B at the same temperature, residual activity decreased until 82% after 48 h, and thereon it remained constant until 144 h of incubation. Similar results were reported with Novozym 435 when incubated in other non-polar [34] and polar solvents [35]; in this case, the enzyme is already immobilized in a moderately hydrophobic support, so it may be that 2M2B does not interact with the enzyme directly due to partition effects [36]. The different behavior of PS-octyl-silica and Novozym 435 denotes that different organic solvents have different capacity to distort the essential water layer around the immobilized lipase because of its polarity [30]. Also other factors, such as the configuration of the enzyme and the type of functional groups in the support in which it was immobilized are determinant of the biocatalyst behavior [35,37,38].

### 2.3. Synthesis of Ascorbyl Palmitate under Standard Conditions

Enzyme activity and substrates solubility play a major role in the esterification of AA, whereby the major challenge is to find out a solvent which can dissolve the substrates at an acceptable concentration while allowing high lipase activity [39]. Synthesis of AsPa was carried out using acetone, acetonitrile and 2M2B at 45 °C and AA:PA molar ratio of 1:5 [20] with both PS octyl-silica and Novozym 435, with the aim of comparing their performance. Synthesis with both biocatalysts using acetone and acetonitrile as reaction media resulted in yields lower than 10% (data non show). On the other hand, higher yield was obtained in 2M2B with Novozym 435, the highest value of 65% being obtained after 144 h of reaction (at that time only 4% was obtained with PS-octyl-silica) (Figure 2). These results can be explained by the intrinsic differences of the biocatalysts and the influence of the reaction conditions on biocatalyst behavior [29]; for example, their stability in the solvents used as reaction media (Table 2). It is necessary then to select the proper reaction conditions for each enzyme considering the immobilization methodology used [8,9,20,24].

The expressed activity of biocatalysts was compared in synthesis and hydrolysis reactions (Table 1 and Table 2). Results show that the expression of these activities run in opposite directions for these biocatalysts: Novozym 435 showed a higher activity of synthesis than PS-octyl-silica, while the opposite occurred for the hydrolytic activity. This indicates that the hydrolytic activity of immobilized lipases does not predict their rate of synthesis, which is probably influenced by the immobilization methodology since diffusional restrictions in the immobilized enzyme will have a different impact according to the properties of the substances diffusing in and out of the biocatalyst [40,41].

In summary, and based on the results of the biocatalyst performances and synthesis yield, Novozym 435 was selected as biocatalyst and 2M2B as solvent, to study the influence of reaction conditions on AsPa synthesis. Novozym 435 has shown good stability properties, being an excellent alternative of immobilized biocatalysts to be used at process conditions [36].

### 2.4. Effect of Reaction Conditions on Ascorbyl Palmitate Synthesis

The enzymatic synthesis enzymatic of AsPa is produced by the lipase catalyzed esterification of AA with PA. In this reaction, the AA:PA molar ratio and the temperature are key variables determining the synthesis yield [20]. An experimental design was conducted considering a temperature, in the range from 41 to 69 °C, and AA:PA molar, in the range from 1:3.8 to 1:12 as variables for the synthesis of AsPa. Table 3 presents the matrix of the experimental design with the values of AsPa yield and the productivity obtained.

The effect of reaction conditions on synthesis yield and the productivity can be clearly perceived in Table 3. For example, synthesis yield increased when the reaction was carried out at higher AA:PA molar ratios at 65 °C. This is probably because the reaction equilibrium was shifted to the ester products, since lipase catalyzed esterification is a kinetically controlled reaction, whereby high concentrations of substrates favor product formation [29]. At similar conditions, Santibáñez, et al. [20] reported lower yields, probably because in that work tert-butanol was used as solvent, negatively affecting enzyme performance. However, the opposite occurred at 45 °C, where low yield was obtained, probably because lower temperatures reduce the substrate diffusion rate inside the biocatalyst particle [8]. Higher AA:PA molar ratios were not evaluated because no increase in the yield of AsPa was attained (experiment 9, Table 3), which may be caused by substrate inhibition [11].

On the other hand, despite that synthesis yield increased with reaction temperature at constant AA:PA molar ratio (experiments 1, 6, Table 3), it reached a maximum at 55 °C and decreased thereon (experiment 7, Table 3). This behavior can be due to the thermal inactivation of the enzyme at higher temperatures [42]; it has been reported that 69 °C is the limit temperature at which Novozym 435 works suitably [26]. Regarding the productivity, it shows similar behavior.

In summary, results show that maximum synthesis yield of 81% was obtained at 55 °C with a AA:PA molar ratio 1:8 (Figure 3), which is higher than reported by other authors [8,20,39].

According to the results previously obtained, the synthesis yield of AsPa varied with the AA:PA molar ratio and temperature; therefore, the synthesis of ascorbyl palmitate with Novozym 435 was optimized in terms of those variables using response surface methodology. This methodology allows determining the interaction between variables and their values that maximize the synthesis yield of AsPa. As a result of the experimental design, the following second-order polynomial equations was obtained for the synthesis yield:(1)$$Y=80.7+6.5\cdot T+5.7\cdot R+8.0\cdot T\cdot R-15.2\cdot T^{2}-13.9\cdot R^{2}-0.17\cdot T^{2}\cdot R-9.25\cdot T\cdot R^{2}$$
where: Y: synthesis yield of AsPa; T: temperature; R: AA:PA molar ratio.

The coefficient of determination for Equation (1) was 0.86. The polynomial equation shows the interaction of both variables and their effect on the synthesis yield, both variables having a strong effect on it. From such equation, optimal operation conditions maximizing the synthesis of AsPa were obtained, being 82% at 57.5 °C and AA:PA molar ratio of 1:9, which is close to the value obtained at the conditions of the central point of the experimental design.

### 2.5. Downstream Processing

The product of the enzymatic reaction of AsPa synthesis was obtained after one batch was subjected to purification. The method used was solvent extraction, as described in the Materials and Methods section. This procedure was established to separate the unreacted substrates, employing solvents that are compatible with the subsequent application of AsPa in foodstuffs or cosmetics. After the filtration step, the purification process consisted in three consecutive steps: vacuum evaporation and solvent extraction in hexane and in ethyl acetate/water mixture (see Material and Methods).

Table 4 presents the recovery yield and product purity obtained. After solvent (2M2B) removal by vacuum evaporation, the purity of the AsPa was 11.4%. The extraction with hexane removed most of the unreacted PA (>99%) from the reaction mixture, while the AA and AsPa remained insoluble in this solvent, obtaining a recovery yield of 94%. Finally, AA was extracted with a mixture of ethyl acetate and water, the AsPa remaining insoluble in the mixture. This last step has a recovery yield of 90% and allowed reaching a final product with a high purity (97.5%). AsPa was successfully purified using a safe and compatible solvent, with a final yield of 84.2%.

The results obtained compare favorably with those of related works, such as in the purification of ascorbyl oleate reported by Viklund, et al. [9], where a yield of only 29% was obtained. The main difference was the use there of diethylether as solvent for AA extraction; in this case, the use of water allowed a better separation of AA, besides being a biocompatible and benign solvent.

There are several publications that report methods of purification or extraction of ascorbyl esters at laboratory scale by using semipreparative HPLC columns [25,26], which requires sophisticated and costly equipment. Other investigations have reported the percentage of purification of the antioxidant, but not the yield of the process [24,27]. The present work considered a thorough analysis of the purification process obtaining values of recovery yield and purity that are significant at a productive scale.

## 3. Materials and Methods 

### 3.1. Materials

Lipase TL from *Pseudomonas stutzeri* (crude powder) was a gift from Meito Sangyo (Tokyo, Japan) Immobilized lipase from *Candida antarctica* (Novozym 435^®^) was kindly donated by Novozym, Spain. Cetyltrimethylammonium bromide (CTAB), trimethoxy(octyl)silane (OTMS; 96%), p-nitrophenol (pNP), p-nitrophenyl butyrate (pNPB) and ascorbyl palmitate were purchased from Sigma-Aldrich (St. Louis, MO, USA). Ascorbic acid (AA) and palmitic acid (PA) were obtained from Loba Chemie (Bombai, India). Sodium silicate (25–29% SiO_2_ and 7.5–9.5% Na_2_O), acetone, acetonitrile, hexane, ethyl acetate, 2-methyl-2-butanol (2M2B) and monopotassium phosphate were of the highest available purity and were purchased from Merck (Darmstadt, Germany).

### 3.2. Synthesis and Functionalization of Silica

Silica synthesis and functionalization were carried out according to Bernal, et al. [43]. The mesoporous support was synthesized with the following molar proportion of reagents SiO_2_:Na_2_O:CTAB:EtAc:H_2_O = 1:0.3:0.24:7.2:193 and heated at 80 °C for 48 h. Then the solid was calcined at 540 °C for 6 h. Silica functionalization began with one gram of previously activated silica (left for 12 h to 150 °C) which was mixed with 30 mL of 5% OTMS in toluene and gently stirred under reflux for 6 h at 105 °C. After filtration, the solid was washed with acetone and abundant 25 mM monopotassium phosphate buffer pH 7.0, and finally dried at room temperature.

### 3.3. Enzymatic Activity Assay

Assay for hydrolytic lipase activity was performed by measuring the increase in absorbance at 348 nm (JASCO ultraviolet (UV)/visible spectrophotometer, Japan) produced by the released p-NP in the hydrolysis of 0.4 mM pNPB in 25 mM monopotassium phosphate buffer at pH 7.0 and 25 °C, using an appropriate amount of matrix-bound biocatalyst or free lipase. One international unit of lipase activity (IU) was defined as the amount of enzyme that hydrolyzes 1 μmol of pNPB per minute under the conditions described above. 

### 3.4. Rate of Reaction of Ascorbyl Palmitate Synthesis

Additionally, the rate of AsPa synthesis of was determined for each immobilized biocatalyst by following the time course of the reaction under standard conditions: 45 °C, 150 rpm, 5 mL 2M2B, 87 mM AA, 1:5 AA:PA molar ratio, 12 g/L of enzyme and 14 g/L of molecular sieve 3A. The specific rate of synthesis was expressed in μmoles of AsPa produced per minute per gram of biocatalyst at the above conditions.

### 3.5. Immobilization of Lipase on Silica Support

Lipase from *P. stutzeri* was dissolved in 25 mM monopotassium phosphate buffer pH 7.0 and then the solution was contacted with the silica support. Immobilization was carried out offering 10 mg protein per g of support. The enzyme-carrier suspension was left under roller agitation at 20 °C until the activity in the supernatant remained constant. The hydrolytic activity of the supernatant and the suspension was monitored during immobilization and measured as indicated in Section 3.3. The biocatalyst was filtered, washed with water and dried at room temperature. Activity was expressed in IU per gram of support. Immobilization process was monitored by measuring the enzymatic activity of samples of the suspension containing the catalyst and of the catalyst-free filtrate at regular time intervals. Immobilization was evaluated in terms of the following parameters:

-Protein immobilization yield (IY_P_), representing the mass (M) percentage of the contacted protein which is immobilized: (2)$${\mathrm{IY}}_{p}=\frac{M_{\mathrm{immobilized}\;\mathrm{protein}}}{M_{\mathrm{contacted}\;\mathrm{protein}}}\cdot 100_{}$$

The mass of immobilized protein was calculated as the difference between the mass of contacted protein and the mass of protein in the filtrate.

-Immobilization yield of expressed activity (IY_a_), representing the percentage of contacted activity (A) that is expressed in the biocatalyst after immobilization:(3)$${\mathrm{IY}}_{a}=\frac{A_{\mathrm{expressed}\;\mathrm{activity}}}{A_{\mathrm{ocontacted}\;\mathrm{activity}}}\cdot 100_{}$$

### 3.6. Stability in the Presence of Solvents

Stability of the enzymes in the different reaction media was determined in order to have an estimate of its behavior under reaction conditions and evaluate the option of biocatalyst reuse. A stability test was performed by measuring the residual activity in time after incubation in the presence of acetone, 2M2B and acetonitrile (all solvents at the equilibrium water concentration, close to 5%), under non-reactive conditions. Incubation was done at constant temperature in a thermoregulated bath under magnetic stirring at 150 rpm at 45 °C. Inactivation of the biocatalysts was monitored by quantifying the hydrolytic activity (see Section 3.3) at regular time intervals. The residual activity was expressed as the activity of the biocatalyst relative to its initial activity (incubation time zero).

### 3.7. Ascorbyl Palmitate Synthesis

Synthesis of AsPa was carried out at 5 mL scale by esterification of AA (87 mM) with palmitic PA in an orbital shaker at 150 rpm. The reaction mixtures contained 2M2B and 12 g/L of enzyme and 14 g/L of molecular sieve 3A used to remove water from the reaction medium. Temperature and molar ratio between substrates were those corresponding to the experimental design (Table 5). Reaction was conducted until reaching the point of maximum AsPa concentration. The assays were carried out in duplicate, with standard deviations never exceeding 5%.

Quantification of substrates and products of synthesis was done as described in Section 3.8. AsPa synthesis was evaluated in terms of synthesis yield (percent):(4)$$Y=\frac{M_{\mathrm{AsPs},\mathrm{final}}}{M_{\mathrm{AA},\mathrm{initial}}}\cdot 100$$
where M_AsPs,final_ represents the final moles of AsPa synthesized and M_AA,initial_ represents the initial moles of the limiting substrate AA.

### 3.8. High-Performance Liquid Chromatography (HPLC) Analysis of the Reaction Products

Substrates and products of synthesis were analyzed in a Jasco RI2031 HPLC equipment, provided with UV-Vis spectrophotometric detector, isocratic pump (JASCO model AS-2089) and autosampler (Jasco AS 2055), using a C-18 column (Análisis Vínicos S.L. Kromasil C18.5 μm 4.6 mm × 150 mm). Separation was achieved by eluting with acetonitrile:water mixtures as mobile phase at a flowrate of 1 mL/min with the following gradient program: 60:40 (*v*/*v*) for 6 min and then 95:5 (*v*/*v*) for 18 min. Retention times of the compounds involved in the synthesis were 1.2, 14.2 and 19.2 min for AA, AsPa and PA, respectively.

### 3.9. Experimental Design

With the purpose of evaluating the effect of reaction conditions on the synthesis of AsPa, a composite central design was carried out considering the synthesis yield as objective function. Temperature and molar ratio between substrates were selected as key variables given their effect on reaction kinetics and enzyme inactivation.

Central composite design of two factors with five replicas at the central point was used to optimize the reaction conditions using response surface methodology. Axial points were determined considering α factor of 1.4142 to obtain a rotatable design. The uncoded and coded values for each level of these variables are given in Table 5. Reactions were conducted as indicated in Section 3.7, but the experimental conditions varied according to the corresponding experimental design. Results were analyzed using the software Design-Expert v10 (Stat-Ease, Minneapolis, MN, USA).

### 3.10. Downstream Processing

Purification of AsPa was done by solvent extraction according to Zhao, et al. [27] with some variations. A schematic representation of the process is in Figure 4. The reaction mixture contained solvent, product, residual substrates, enzyme and molecular sieve. The enzyme and the molecular sieve were retained by a 0.16 mm pore filter paper. The solvent was removed by evaporation on a R-100 Büchi Rotavapor for 1 h at 40 °C under vacuum (V-100Vacuum Pump Büchi). The solid containing the product and residual substrates was recovered and PA was separated from the mixture by four extractions with a total of 40 mL of hexane. At the end of each extraction step, the solid was separated from the hexane by centrifugation at 19,460 *g* for 20 min at room temperature. Finally, AA was extracted from the resulting solid with a mixture of ethyl acetate (0.5 mL) and water (10 mL) once, and twice with an additional 20 mL of water. The liquid and the solid were separated by centrifugation at the same conditions as in the previous procedure.

The process of purification of AsPa was evaluated in terms of purity (P) and product recovery yield (RY), using the following equations:(5)$$P\;\left(\%\right)=\frac{M_{F\;\mathrm{AsPa}}}{M_{\mathrm{FT}}}\cdot 100$$
(6)$$\mathrm{RY}\;\left(\%\right)=\frac{M_{F\;\mathrm{AsPa}}}{M_{I\;\mathrm{AsPa}}}\cdot 100$$
where M_I AsPa_ and M_F AsPa_ are the mass of AsPa at the beginning and at the end of each purification step respectively, and M_FT_ is the total mass recovered at the end of each purification step.

## 4. Conclusions

*Pseudomonas stutzeri* lipase immobilized on silica octyl and a commercial enzyme Novozym 435 were evaluated as catalysts for the synthesis of AsPa. Novozym 435 proved to be the most appropriate in this case, being superior in terms of the rate of synthesis of AsPa and stability in the presence of the solvent 2M2B, where the solubility of AA was higher. Temperature and AA:PA molar ratio were studied as variables, both being significant, having a strong effect on the yield of AsPa synthesis. The highest AsPa yield was attained in 2M2B at 55 °C and AA:PA molar ratio 1:8.

The product was purified by solvent extraction, obtaining a purity of 97.5% with a recovery yield of 84%. The process considering the enzymatic synthesis of AsPa and its purification by solvent extraction is a contribution towards a greener production of AsPa intended for its use as an additive in food and cosmetic products. 

## Figures and Tables

**Figure 1 molecules-24-03227-f001:**
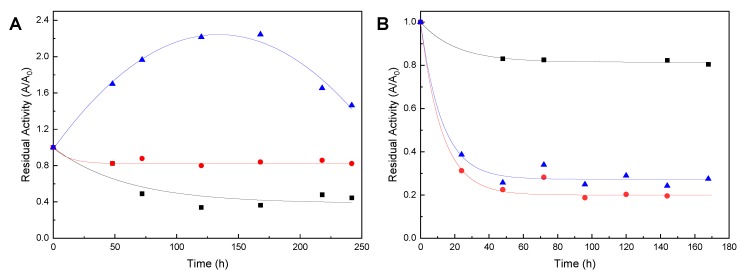
Stability of biocatalysts in organic solvents under non-reactive conditions at 45 °C. (**A**) Lipase from *P. stutzeri* immobilized in octyl-silica; (**B**) Novozym 435. Acetonitrile (blue triangles), acetone (red circles) and 2M2B (black squares).

**Figure 2 molecules-24-03227-f002:**
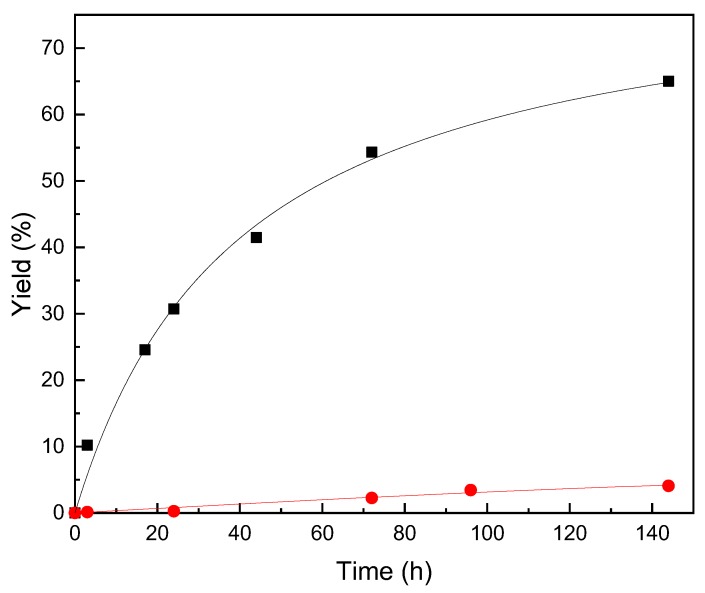
Synthesis of ascorbyl palmitate with Novozym 435 (black squares) and PS-octyl-silica (red circles). Reaction conditions: 87 mM of AA, 1:5 AA:PA molar ratio, 5 mL of 2M2B, 12 g/L of enzyme and 14 g/L of molecular sieve 3A, at 45 °C and 150 rpm.

**Figure 3 molecules-24-03227-f003:**
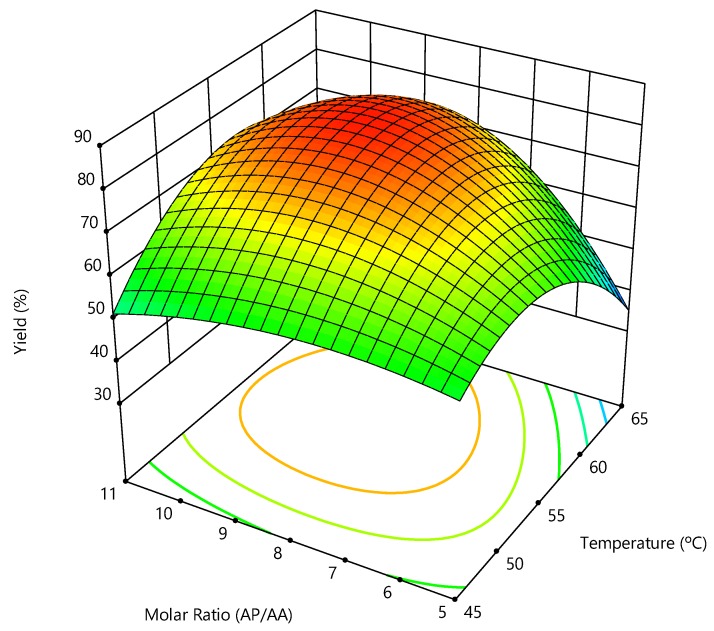
Response surface for the effect of temperature and ascorbic acid: palmitic acid molar ratio on the synthesis yield of ascorbyl palmitate. The synthesis reaction was performed in pure 2M2B at 150 rpm with 12 g/L of Novozym 435.

**Figure 4 molecules-24-03227-f004:**
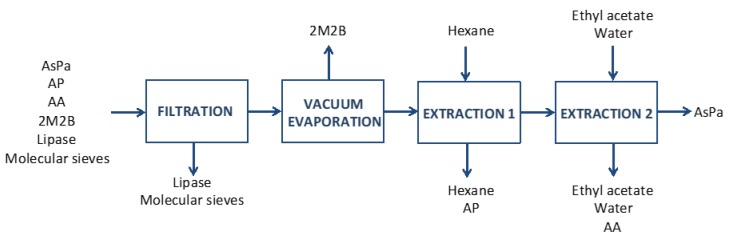
Schematic representation of the downstream process of ascorbyl palmitate synthesis.

**Table 1 molecules-24-03227-t001:** Characterization of lipase biocatalysts.

Biocatalyst	Hydrolytic Activity	Residual Activity (A/A_0_) ^1^		
	IU/g_biocatalyst_	Acetone	Acetonitrile	2M2B
PS-octyl-silica	70.44	0.84	2.24	0.36
Novozym 435	34.61	0.20	0.24	0.82

^1^ After incubation for 144 h.

**Table 2 molecules-24-03227-t002:** Summary of parameters of synthesis of ascorbyl palmitate (AsPa) with the immobilized biocatalysts.

Biocatalysts	Time	Initial Rate	Yield	Productivity
	(h)	(μmol × min^−1^ × mL^−1^)	(%)	(mg_AsPa_ × g_biocat_^−1^ × h^−1^)
Novozym 435	144	411.0	65	13.6
PS octyl-silica	144	3.7	4.1	0.9

**Table 3 molecules-24-03227-t003:** Experimental design and results obtained in the reaction of synthesis of ascorbyl palmitate catalyzed by Novozym 435, using 2M2B as a solvent after 144 h of reaction, considering temperature and ascorbic acid to palmitic acid molar ratio (AA:AP) as variables.

Experiment	Temperature (°C)	AA:PA Molar Ratio	Yield (%)	Productivity (mg_AsPa_ × g_cat_^−1^ × h^−1^)
1	55	1:8	81 ± 2	16.8
2	45	1:5	65 ± 2	13.6
3	45	1:11	60 ± 2	12.5
4	65	1:5	43 ± 2	9.1
5	65	1:11	71 ± 2	14.7
6	41	1:8	33 ± 2	6.9
7	69	1:8	51 ± 2	10.7
8	55	1:3.8	37 ± 0	7.7
9	55	1:12.2	53 ± 0	11.0

**Table 4 molecules-24-03227-t004:** Composition, recovery yield and purity of ascorbyl palmitate (AsPa) at each step of the purification process.

Component	Vacuum Evaporation	Extraction 1	Extraction 2
AsPa (g)	1.39	1.3	1.17
PA (g)	10.4	0.09	–
AA (g)	0.5	0.5	0.07
2M2B (g)	–	–	–
Total (g)	12.2	1.9	1.2
**Recovery Yield (%)**	**100**	**93.5**	**90.0**
**Purity (%)**	**11.4**	**68.4**	**97.5**

**Table 5 molecules-24-03227-t005:** Variables and levels selected of the synthesis of ascorbyl palmitate evaluated for a composite central design.

Variable	Levels				
	−1.41	−1	0	1	1.41
Temperature (°C)	41	45	55	65	69
Molar ratio AA:PA	1:3.8	1:5	1:8	1:11	1:12.2

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
