# Peer review of "Synthesis with Immobilized Lipases and Downstream Processing of Ascorbyl Palmitate"

_molecules, 2019, doi:10.3390/molecules24183227_

Round 1
Reviewer 1 Report
The authors performed a very interesting study using a non-commercial biocatalyst to synthesize a cometic ester The paper evaluates the influence of the relevant factors on the ester synthesis. The purpose of the investigation was also discussed in satisfactory manner showing a good theoretical background. It is important to note that the scope of the work is a very important subject under intensive investigation and is valuable for its contribution to increase in the possibility of using enzymes in industrial processes. Thus, the manuscript is suitable for publication, as it is.
Author Response
Reviewer 1.
Comments and Suggestions for Authors
The authors performed a very interesting study using a non-commercial biocatalyst to synthesize a cometic ester The paper evaluates the influence of the relevant factors on the ester synthesis. The purpose of the investigation was also discussed in satisfactory manner showing a good theoretical background. It is important to note that the scope of the work is a very important subject under intensive investigation and is valuable for its contribution to increase in the possibility of using enzymes in industrial processes. Thus, the manuscript is suitable for publication, as it is.
Reviewer 2 Report
The paper shows an interesting process catalyzed by a commercially available lipase, comparing the results with a home-made biocatalysts. I have doubts that the home-made catalyst give some relevant data on the process, as the commercial one is much better and they use two supports and two different enzymes, that way is difficult to know if the difference is by the enzyme or by the immobilization protocol. Perhaps they can focus just on the use of Novozyme 435 on this process, or use more the homemade biocatalysts in dicussion to justify.
Specifci points are listed below
Title: define the enzyme utilized in this paper
Immobilization is almost not treated in introduction. A proper immobilization can improve enzyme activity, selectivity, specificity, stability, resistance to inhibitors and chemicals, even purity. There are reviews on each of these aspects, and any of them are included.
Interfacial activation of lipases needs also to be discussed, as well as immobilization of lipases on hydrophobic supports. This review may be of interest:
ScopusEXPORT DATE:20 Aug 2019 Rodrigues, R.C., Virgen-Ortíz, J.J., dos Santos, J.C.S., Berenguer-Murcia, Á., Alcantara, A.R., Barbosa, O., Ortiz, C., Fernandez-Lafuente, R.24339451300;55200320500;56962733600;6508313670;7006761900;37053343800;35556523500;35552449900;Immobilization of lipases on hydrophobic supports: immobilization mechanism, advantages, problems, and solutions(2019) Biotechnology Advances, 37 (5), pp. 746-770. Cited 9 times.https://www.scopus.com/inward/record.uri?eid=2-s2.0-85064038560&doi=10.1016%2fj.biotechadv.2019.04.003&partnerID=40&md5=d5e11794e3230efa485d077dca7fe416 DOI: 10.1016/j.biotechadv.2019.04.003DOCUMENT TYPE: ReviewPUBLICATION STAGE: FinalSOURCE: Scopus
The enzyme should be better introduced, this review may be also of interest.
Rios, N.S., Pinheiro, B.B., Pinheiro, M.P., Bezerra, R.M., dos Santos, J.C.S., Barros Gonçalves, L.R.
57191074032;57196008187;57209112304;57194461003;56962733600;7103169186;
Biotechnological potential of lipases from Pseudomonas: Sources, properties and applications
(2018) Process Biochemistry, 75, pp. 99-120. Cited 9 times.
https://www.scopus.com/inward/record.uri?eid=2-s2.0-85052996408&doi=10.1016%2fj.procbio.2018.09.003&partnerID=40&md5=396f81be2d804d2bad3a4909ff8f1884
DOI: 10.1016/j.procbio.2018.09.003
DOCUMENT TYPE: Review
PUBLICATION STAGE: Final
SOURCE: Scopus
The synthetic strategy should be also introduced: esterification (thermodynamically controlled synthesis), where the main problem is water accumulation. Marty has several papers in this problem, that may be solved using very hydrophobic supports or ultrasounds (see Rodrigues).
“Hydrolytic specific activity of PS-octyl-silica was 70.44 100 IU/gbiocatalyst; this activity was higher than reported for Pseudomonas stutzeri lipase immobilized in 101 octadecyl-Sepabeads [23], probably because hydrophobicity of octyl-silica support confers the 102 enzyme an environment prone to interfacial activation [24].” Did they suggest that this is not the case using octadecyl-Sepabeads?? Why? They should give the amount of enzyme immobilized in both cases and compare specific activity, not only mass activity (moreover, it is dried or wet)? Specific area (that determine the loading), pores diameter (that determine substrate diffusion) of the support may easily explain the results. Mention the substrate used in this point.
Hyperactivation may be derived from enzyme release from the support?
“Since the support is hydrophobically activated” Really, it is hydrophobic, but this suggests some chemical reactive group.
“in this case, the enzyme is already immobilized in a 132 highly hydrophobic support,” Lewatic is not so hydrophobic.
Author Response
Reviewer 2.
Comments and Suggestions for Authors
The paper shows an interesting process catalyzed by a commercially available lipase, comparing the results with a home-made biocatalysts. I have doubts that the home-made catalyst give some relevant data on the process, as the commercial one is much better and they use two supports and two different enzymes, that way is difficult to know if the difference is by the enzyme or by the immobilization protocol. Perhaps they can focus just on the use of Novozyme 435 on this process, or use more the homemade biocatalysts in discussion to justify.
Answer: Novozym 435 is usually the best choice for conducting this type of reactions, so the original purpose was to challenge a home-made lipase. Results confirm the supremacy of Novozym 435, but we feel that effort in trying to use cheaper enzymes is valid. We cannot ascertain if the difference is due to the nature of the enzyme or the immobilization procedure, so we can only guess that this is due to both, even though good results have been obtained in other cases with lipases immobilized in hydrophobic silica derivatives.
Specific points are listed below
-Title: define the enzyme utilized in this paper
Answer: The enzyme used have been inserted in the title. The new title is: Synthesis With Immobilized Lipases And Downstream Processing Of Ascorbyl Palmitate. (Line 2).
-Reviewer: Immobilization is almost not treated in introduction. A proper immobilization can improve enzyme activity, selectivity, specificity, stability, resistance to inhibitors and chemicals, even purity. There are reviews on each of these aspects, and any of them are included.
Interfacial activation of lipases needs also to be discussed, as well as immobilization of lipases on hydrophobic supports. This review may be of interest: Rodrigues, R.C., Virgen-Ortíz, J.J., dos Santos, J.C.S., Berenguer-Murcia, Á., Alcantara, A.R., Barbosa, O., Ortiz, C., Fernandez-Lafuente, R. Immobilization of lipases on hydrophobic supports: immobilization mechanism, advantages, problems, and solutions (2019) Biotechnology Advances, 37 (5), pp. 746-770.
Answer: A new paragraph has been introduced describing the lipase immobilization (Lines 57-60). The suggested reference and others have been included.
Reviewer: The enzyme should be better introduced, this review may be also of interest. Rios, N.S., Pinheiro, B.B., Pinheiro, M.P., Bezerra, R.M., dos Santos, J.C.S., Barros Gonçalves, L.R. Biotechnological potential of lipases from Pseudomonas: Sources, properties and applications. (2018) Process Biochemistry, 75, pp. 99-120.
Answer: A new sentence has been introduced to describe the enzyme. The suggested reference has been included (Lines 77-78).
Reviewer: The synthetic strategy should be also introduced: esterification (thermodynamically controlled synthesis), where the main problem is water accumulation. Marty has several papers in this problem, that may be solved using very hydrophobic supports or ultrasounds (see Rodrigues).
Answer: The strategy of ascorbyl palmitate synthesis has been introduced. The suggested references and others have been included (Lines 50-56).
Reviewer: “Hydrolytic specific activity of PS-octyl-silica was 70.44 100 IU/gbiocatalyst; this activity was higher than reported for Pseudomonas stutzeri lipase immobilized in 101 octadecyl-Sepabeads [23], probably because hydrophobicity of octyl-silica support confers the 102 enzyme an environment prone to interfacial activation [24].” Did they suggest that this is not the case using octadecyl-Sepabeads?? Why? They should give the amount of enzyme immobilized in both cases and compare specific activity, not only mass activity (moreover, it is dried or wet)? Specific area (that determine the loading), pores diameter (that determine substrate diffusion) of the support may easily explain the results. Mention the substrate used in this point.
Answer: The paragraph has been deleted. The comparison between both immobilized enzymes was indeed rather meaningless because of the quite different conditions used.
Reviewer: Hyperactivation may be derived from enzyme release from the support_?
Answer: The hyperactivation of the PS-octyl-silica in the presence of acetonitrile (Figure 1) is not due to the release of the enzyme from the support. The protein content on the support remained constant during the incubation in solvents (data not shown). Moreover, evidence of hyperactivation of immobilized lipases on hydrophobic support has been already reported (Quilles, J.C.J.; Brito, R.R.; Borges, J.P.; Aragon, C.C.; Fernandez-Lorente, G.; Bocchini-Martins, D.A.; Gomes, E.; da Silva, R.; Boscolo, M.; Guisan, J.M. Modulation of the activity and selectivity of the immobilized lipases by surfactants and solvents. Biochemical Engineering Journal 2015, 93, 274-280, doi:10.1016/j.bej.2014.10.009.). A new sentence and the mentioned reference have been included [31] (Line 130-131)
Reviewer: “Since the support is hydrophobically activated” Really, it is hydrophobic, but this suggests some chemical reactive group.
Answer: This sentence refers to the octyl-silica support, which is described in the hydrophobically functionalized with OTMS (Octyltrimethoxysilane), according to the methodology described in Section 4.2. The sentence has been rewritten to make this point clearer. (Lines 134-136)
Reviewer: “in this case, the enzyme is already immobilized in a 132 highly hydrophobic support,” Lewatic is not so hydrophobic.
Answer: As the reviewer points out, Lewatit support is not so hydrophobic as other immobilization support like Sepabeads and Purolite, but much more hydrophobic than mesoporous silica used as support in this work. The sentence has been rewritten according to the reviewer’s comment. (Lines 142-144)
Reviewer 3 Report
Please find my comments about the results and the presentation below:
Line 10: there is additional email address. Authors must review the text and correct the names of the microorganisms. They should be in italic font. Lines 65-68. The sentence is too long. Please divide the sentence into two. Line125: There are literature references that are not in the bibliography. In table 4: please enter in the table the name of the compound for which purity was determined. Line 259: there are wrong names of compounds. Figure 4: what mean “?” The authors must corrected it.Author Response
Reviewer 3
Comments and Suggestions for Authors
Please find my comments about the results and the presentation below:
Reviewer: - Line 10: there is additional email address.
Answer: The additional email address was deleted (Line 10) .
Reviewer: - Authors must review the text and correct the names of the microorganisms. They should be in italic font.
Answer: All microorganisms’ names are now in italics.
Reviewer: Lines 65-68. The sentence is too long. Please divide the sentence into two.
Answer: The sentence has been reformulated and divided into two. (Lines 75-80)
Reviewer: Line125: There are literature references that are not in the bibliography.
Answer: References [32,33] have been included in the bibliography. (Line 136)
Reviewer: In table 4: please enter in the table the name of the compound for which purity was determined.
Answer: The table caption has been changed to include the name of ascorbyl palmitate (AsPa). (Line 243)
Reviewer: Line 259: there are wrong names of compounds.
Answer: The names of the compounds have been corrected (Lines 268)
Reviewer Figure 4: what mean “?” The authors must corrected it.
Answer: There is no sign “?” in Figure 4 (Lines 360).
Round 2
Reviewer 2 Report
The paper has been corrected